# Bacterial Adhesion of TESPSA and Citric Acid on Different Titanium Surfaces Substrate Roughness: An In Vitro Study with a Multispecies Oral Biofilm Model

**DOI:** 10.3390/ma16134592

**Published:** 2023-06-25

**Authors:** Javi Vilarrasa, Gerard Àlvarez, Agnès Soler-Ollé, Javier Gil, José Nart, Vanessa Blanc

**Affiliations:** 1Department of Periodontology, Universitat Internacional de Catalunya, 08195 Barcelona, Spain; jvilarrasa@uic.es (J.V.); jose.nart@uic.es (J.N.); 2Department of Microbiology, DENTAID Research Center, 08290 Cerdanyola del Vallès, Spain; agnes.soler@dentaid.es (A.S.-O.); blanc@dentaid.es (V.B.); 3Bioengineering Institute of Technology, Universitat Internacional de Catalunya, 08195 Sant Cugat del Vallés, Spain

**Keywords:** titanium, peri-implantitis, biomaterials, TESPSA, citric acid

## Abstract

This in vitro study analyzed the influence of substrate roughness on biofilm adhesion and cellular viability over triethoxysilylpropyl succinic anhydride silane (TESPSA)- and citric acid (CA)-coated surfaces at 12 and 24 h, respectively. A multispecies biofilm composed of *S. oralis*, *A. naslundii*, *V. parvula*, *F. nucleatum*, *P. intermedia*, *P. gingivalis*, *P. endodontalis* and *F. alocis* was developed over titanium discs grouped depending on their roughness (low, medium, high) and antibacterial coating (low-TESPSA, medium-TESPSA, high-TESPSA, and CA). The biofilm was quantified by means of quantitative polymerase chain reaction (PCR) and viability PCR and assessed through confocal laser scanning microscope (CLSM). Quantitative PCR revealed no significant differences in bacterial adhesion and biofilm mortality. CA was the surface with the lowest bacterial counts and highest mortality at 12 and 24 h, respectively, while high harbored the highest amount of biofilm at 24 h. By CLSM, CA presented significant amounts of dead cells compared to medium-TESPSA and high-TESPSA. A significantly greater volume of dead cells was found at 12 h in low-TESPSA compared to medium-TESPSA, while CA also presented significant amounts of dead cells compared to medium-TESPSA and high-TESPSA. With regard to the live/dead ratio, low-TESPSA presented a significantly higher ratio at 12 h compared to medium-TESPSA and high-TESPSA. Similarly, CA exhibited a significantly higher live/dead ratio compared to medium-TESPSA and high-TESPSA at 12 h. This multispecies in vitro biofilm did not evidence clear antiadhesive and bactericidal differences between surfaces, although a tendency to reduce adhesion and increase antibacterial effect was observed in the low-TESPSA and CA.

## 1. Introduction

One of the major clinical challenges in the contemporary implant dentistry is the prevention and management of peri-implant diseases. In the 2017 World Workshop in Periodontology, peri-implant mucositis was defined as an inflammatory lesion confined to the soft tissue surrounding dental implants, while peri-implantitis involved a subsequent progressive loss of supporting bone [1]. Recent epidemiologic research [2,3] reported that peri-implant mucositis and peri-implantitis occur in almost 25% of patients. In other words, peri-implant diseases are common infectious diseases among people with dental implants.

Peri-implant diseases are caused by site-specific infections associated with a heterogeneous, complex microbial community [4,5]. As a matter of fact, the accumulation of biofilm around dental implants is one of the risk factors most associated with peri-implant disease [6]. Peri-implantitis microbiota encompasses traditional periodontopathic microorganisms as well as new pathogens identified with novel technologies such as 16S metagenomic sequencing. The core peri-implant microbiome seems to be enriched with *Fusobacterium*, *Parvimonas* and *Campylobacter* species [7]. Moreover, *Treponema denticola* and high levels of *Porphyromonas gingivalis*, as well as other newly discovered species, including *Filifactor alocis*, *Fretibacterium fastidiosum* and *Treponema maltophilum*, have been recognized in peri-implantitis sites [7].

Over the decades, clinical scientific research has promoted a substantial improvement of design and surface topography and a superior understanding of bone and soft tissue biology. Initially, implant surface modifications were developed to positively modulate the host-to-implant tissue response [8]. In this sense, it is important to underline that rougher surfaces have led to an improved bone-implant response [9], thus increasing implant survival rates and predictability in challenging scenarios [10]. On the contrary, implant surface properties may also play a pivotal role in initial bacterial adhesion as long as microorganisms colonize titanium surfaces early after implant placement [11]. Accordingly, greater surface roughness may favor bacterial adhesion [12,13], mainly due to the larger contact area between the material surface and the bacterial cells [14], and to the protection of shear forces [15]. Nonetheless, the clinical impact of implant surface on long-term peri-implant bone stability or peri-implant health remains currently uncertain and controversial [10]. Over recent years, promising preclinical and clinical research has focused on the development of modified surfaces with the aim of enhancing soft tissue healing and reducing biofilm overgrowth [16,17]. Moreover, some research has sought to discover antibacterial solutions to further reduce bacterial viability [18]. In this context, our research group has focused on triethoxysilylpropyl succinic anhydride silane (TESPSA), which is an anchoring platform that synergizes with attached biomolecules to enhance osteoinductive and antibacterial surface properties [19,20,21], and on citric acid passivation, which provides a contrasted bactericidal effect [22,23]. The silanes compounds hydrolyze into multiple silanol groups, producing a hydroxylsilane. This molecule can create a covalent bond with the hydroxyls present on titanium surfaces and fix the hydroxilsilane to the metal surface [24]. Moreover, it is able to covalently immobilize biomolecules, promoting specific cell responses [25]. Thus, the silane acts as a binder between the substrate and the biomolecules [26]. However, the triethoxysilylpropyl succinic anhydride (TESPSA) presents antibacterial behavior. This behavior occurs without the need to bind antibacterial biomolecules such as lactoferrine, since it produces an oxidation of the bacteria in contact with them.

CA at low pH is able to freely cross the microbial membrane, and once inside the cytoplasm, it dissociates into CA anions and protons, leading to the acidification of the intracellular media—thus causing functional and structural damage to the bacteria [27].

The influence of substrate surface roughness on the antiadhesive and antibacterial capacity of these coatings needs to be further studied. Therefore, the main objectives of this in vitro study were to analyze the influence of substrate roughness on biofilm adhesion and to quantify early bacterial adhesion and cellular viability over TESPSA and citric acid surfaces.

The alternative hypothesis was that antibacterial-coated surfaces would have a greater antibacterial effect in machined substrates than in rough ones.

## 2. Materials and Methods

The flowchart of the study is represented in Figure 1.

### 2.1. Study Groups

Titanium discs (5 mm in diameter and 2 mm in width) with different surface treatments were provided by SOADCO S.L. (Klockner Dental Implant, Esclades Engordany, Andorra). Commercially pure titanium was of grade 3 (Ti: 99.5%, O: 0.3%, Fe: 0.1%, C: 0.05%, N:0.05%). Surface characterization of the samples was performed at the Department of Biomaterials, Biomechanics and Tissue Engineering at Universitat Politècnica de Catalunya (UPC). The roughness was obtained by spraying aluminum oxide (Al_2_O_3_) abrasive particles on the titanium surface at a pressure of 2.5 bars and a gun-sample separation of 70 mm. The low, medium and high roughness were achieved by using aluminum abrasive particles of 10, 30 and 45 μm, respectively. The evaluation of surface roughness was performed by means of confocal laser scanning microscopy (CLSM; OLS Olympus Lext 3000, Shinjuku, Japan). First, the equipment was verified with the use of a reference sample (Mitutoyo SR 15, Elgoibar, Spain “Precision Reference Specimen”: Ra = 0.43 μm). A total of 3 measurements in 3 samples of each surface were calculated. The surfaces observed by scanning electron microcopy are shown in Figure 2. The contact angle analysis was performed with ultrapure distilled water (Millipore Milli-Q, Merck Millipore Corporation, Darmstadt, Germany) and formamide (Contact Angle System OCA15plus—Dataphysics, Filderstadt, Germany) and the corresponding data were analyzed with SCA20 (Dataphysics, Filderstadt, Germany). Contact angle measurements were made with the sessile drop method. Drops were generated with a micrometric syringe and were deposited over discs. A total of 3 μL of distilled water and 1 μL of formamide were deposited on each sample at 200 μL/min. Finally, surface energy was determined by applying the Owens, Wendt, Rabel and Kaelble (OWRK) equation to the wettability values obtained with the distilled water and formamide.

Depending on their roughness (low, medium or high) and antibacterial coating, discs were divided into 7 different surfaces: L, L-TESPSA, M, M-TESPSA, H, H-TESPSA, and citric acid passivated (CA). TESPSA silanization and citric acid passivation over titanium samples were performed as described elsewhere [20,22]. (CLSM images of the different surfaces studied can be observed in Appendix A).

### 2.2. Bacterial Strains and Growth Conditions

The biofilm was composed of eight bacterial species: Streptococcus oralis, Actinomyces naeslundii, Veillonella parvula, Fusobacterium nucleatum, Prevotella intermedia, Porphyromonas gingivalis, Porphyromonas endodontalis and Filifactor alocis. All species were grown on non-selective blood agar plates (No. 2 of Oxoid; Oxoid Ltd., Heisham, Ireland) with 5% defibrinated horse blood, hemin (5 mg/L) and menadione (1 mg/mL). Liquid cultures were prepared in modified brain heart infusion (pH 7.50) [28]. For Filifactor alocis, this medium was set at pH 7.55 and included 0.2% arginine and 5% fetal bovine serum, but neither mucin nor glutamic acid. All species were grown anaerobically at 37 °C.

### 2.3. Biofilm Formation

Biofilms were formed on titanium discs within the wells of pre-sterilized polystyrene 24-well cell culture plates (Greiner Bio-one, Kremsmünster, Austria). A total of twelve replicates were used for each surface. Each disc was immersed in 1.5 mL of a bacterial suspension containing 10^5^ CFU/mL of each bacterial species, except for *S. oralis* (10^4^), *P. endodontalis* (10^6^) and *F. alocis* (10^6^). This bacterial suspension was prepared by diluting overnight pure cultures of each species in fresh modified BHI [25]. The samples were incubated at 37 °C in anaerobic conditions for 12 and 24 h. Thereafter, discs were rinsed by immersion with PBS for 3 s and prepared for different analytical methods. A total of 7 discs were used for qPCR and vPCR analyses, while 3 discs were used for CLSM and 2 were additional discs in case needed.

### 2.4. Propidium Monoazide (PMA) Treatment

Each rinsed disc was placed in a 1.5 mL microcentrifuge tube containing 1 mL of sterile PBS, and the tubes were vortexed for 5 min in order to detach the biofilm. From each sample, 250 μL was transferred into qPCR- and vPCR-labelled tubes. qPCR tubes were intended for quantitative PCR (qPCR), which measures live and dead bacteria, and vPCR tubes for viability quantitative PCR (vPCR), which quantifies live bacteria thanks to a PMA treatment prior to the qPCR. The treatment with PMA was conducted as previously described [25], but with some modifications: vPCR tubes were incubated for 5 min in the dark with 50 µM PMAxx dye (Biotium, San Francisco, CA, USA) and light-exposed for 3 min using a 650 W halogen light source. Thereafter, both qPCR and vPCR tubes were pelleted at 12,000 rpm for 4 min and stored at −80 °C.

### 2.5. DNA Extraction and qPCR

DNA from the bacterial cell pellet was purified using the QlAamp DNA Mini Kit (Qiagen, Hilden, Germany). Briefly, cells were suspended in 180 μL of a 20 mg/mL lysozyme solution (20 mM Tris-HCl, pH 8.0; 2 mM EDTA; 1.2% Triton X-100) and incubated for 30 min at 37 °C. Then, 200 μL of Buffer AL, 15 μL of 20 mg/mL proteinase K and 15 μL of 20 mg/mL RNase A were added and incubated for 15 min at 70 °C. Further procedure was conducted according to the manufacturer’s protocol. Lastly, DNA was eluted in 50 μL of buffer AE. qPCR was conducted by means of LightCycler 480 II (Roche Diagnostics, Basel, Switzerland), using universal oligonucleotides and a universal probe (Life Technologies, Thermo Fisher Scientific, Waltham, MA, USA) for the quantification of all bacteria (Nadkarni et al., 2002). Reactions containing 0.6 µM forward primer, 0.7 µM reverse primer and a 0.2 µM probe were prepared with LightCycler^®^ 480 Probes Master (Roche, Basel, Switzerland) following the manufacturer’s instructions. The standard curve was prepared using DNA from *S. gordonii* ATCC 49818, *V. parvula* NCTC 11810, *A. naeslundii* DSM 17233, *F. nucleatum* DSM 20482 and *P. gingivalis* DSM 20709 as templates [25].

### 2.6. Confocal Laser Scanning Microscope (CLSM)

CLSM was used to evaluate cell viability. Biofilms grown on the titanium discs were stained with LIVE/DEAD^®^ BacLightTM Bacterial Viability Kit (Invitrogen, Waltham, MA, USA) by incubation with 50 μL of the dye solution for 15 min at room temperature. Images were acquired with LEICA TCS SP5 X CLSM using LAS AF^®^ software (Leica Microsystems CMS GmbH, Wetzlar, Germany) at three random positions on the surfaces with a 25× objective. In order to calculate cell viability within the biofilm, confocal micrographs were analyzed using LEICA MM AF software powered by MetaMorph (Molecular Devices, Inc., San Jose, CA, USA) by a single independent operator.

### 2.7. Statistical Analysis

The number of total bacterial cells (qPCR) and live bacterial cells (vPCR) for each surface were transformed to log_10_ to symmetrize data distribution. Biofilm mortality was calculated through the formula log_10_(qPCR/vPCR), which is the ratio between the total cells (live and dead) and the live cells, and therefore roughly the proportion of dead/live cells on a log_10_ scale.

The normal distribution of the data was calculated using the Shapiro–Wilk test. If the data followed a normal distribution, the Student’s *t*-test or ANOVA was used to compare the amount of total bacterial or cell mortality between 2 or >2 study groups, respectively. Otherwise, the U-Mann–Whitney test was performed when comparing two independent groups and the Kruskal–Wallis equality-of-populations rank test to compare >2 independent groups. If necessary, Benjamini–Hochberg corrections were performed to minimize type I errors. All statistical analyses were performed using Stata^®^ (v 15.1). A *p*-value of <0.05 was considered as statistically significant.

## 3. Results

### 3.1. Characterization of the Titanium Surfaces

The main surface properties (i.e., roughness, contact angle and surface energy) of the study groups are summarized in Table 1.

### 3.2. qPCR

#### 3.2.1. Influence of Surface Topography on Bacterial Adhesion

The total bacterial load adhered to each surface was measured by qPCR (Figure 3). CA showed the lowest bacterial counts at 12 h. In addition, the roughest surface (i.e., H) was the condition most coated by bacterial biofilm at 24 h. However, no statistically significant differences were seen between surfaces at either 12 (*p* = 0.161) or 24 h (*p* = 0.194).

#### 3.2.2. Influence of Substrate Roughness of Coated Surfaces on Bacterial Adhesion

The impact of surface roughness on bacterial adhesion was further analyzed only among surfaces coated with TESPSA and CA (Figure 3). Briefly, it was observed that antibacterial surfaces significantly differed at 12 h (*p* = 0.047). H-TESPSA, which is the surface with the greatest roughness and one of the highest bacterial counts at 12 h, exhibited statistically significant differences when compared to CA (*p* = 0.02) (Table 2), which showed the lowest bacterial counts. Furthermore, if a probability of error of 10% had been accepted, the differences between H-TESPSA and L-TESPSA (*p* = 0.06) and between H-TESPSA and M-TESPSA (*p* = 0.07) would also have been significant. CA was the condition most coated by bacterial biofilm at 24 h. However, no significant differences were observed in the total biofilm load when comparing the different substrate roughness of antibacterial surfaces at 24 h (*p* = 0.139). (Appendix A, show the total bacterial in relation to the different surfaces)

#### 3.2.3. Influence of Antibacterial Coating on Bacterial Proliferation

The antibacterial coating effect was studied through the difference in the total number of bacteria between 12 and 24 h on all surfaces (Figure 3). CA was the only condition that showed significant growth in the biofilm biomass from 12 to 24 h (*p* = 0.001), while a significant decrease was shown in the condition of M-TESPSA (*p* = 0.025). When comparing 12–24 h of total biofilm growth among surfaces, there were significant inter-group differences (*p* = 0.014). A post hoc analysis revealed that total biofilm growth was significantly higher in CA compared to surface L-TESPSA (*p* = 0.046) and M (*p* = 0.040).

#### 3.2.4. Influence of Antibacterial Coating on Biofilm Mortality

The overall mortality estimation among different surfaces at 12 and 24 h is depicted in Table 3. At 12 and 24 h, CA was the surface that presented the highest biofilm mortality. However, no statistically significant differences were observed between surfaces at 12 (*p* = 0.53) (Appendix A) and 24 h (*p* = 0.715) (Appendix A). There was only one condition (i.e., H-TESPSA) that experienced a significant reduction in biofilm mortality between 12 and 24 h (*p* = 0.04).

### 3.3. CLSM

CLSM images from the biofilm formed on every single surface at 12 and 24 h are shown in Figure 4 and Figure 5.

#### 3.3.1. Influence of Antibacterial Coating on Total Biofilm Area

Average biofilm area (μm^2^) occupied by total, live and dead bacteria was compared among the different surfaces at 12 and 24 h (Table 4 and Appendix A, ). From all comparisons, L-TESPSA and CA harbored a significantly greater area of dead cells when compared to M-TESPSA (*p* = 0.01 and 0.02, respectively) and H-TESPSA (*p* = 0.02 in both cases) at 12 h.

#### 3.3.2. Influence of Antibacterial Coating on Total Biofilm Volume

Average biofilm volume (μm^3^) occupied by total, live and dead bacteria was compared among the different surfaces at 12 and 24 h (Table 5) (Appendix A
*p*-values of comparison of live/dead bacteria in relation to the surfaces). Among all comparisons, there was a significantly greater volume of dead cells at 12 h in L-TESPSA when compared to M-TESPSA (*p* = 0.005) and H-TESPSA (*p* = 0.20), while CA also presented significant amounts of dead cells compared to M-TESPSA (*p* = 0.01) and H-TESPSA (*p* = 0.02).

#### 3.3.3. Influence of Antibacterial Coating on Live/Dead Ratio

The live/dead ratio from each surface at 12 and 24 h is summarized in Table 6. At 12 h, L-TESPSA and CA presented a significantly lower live/dead ratio compared to M-TESPSA (*p* = 0.01 in both comparisons) and H-TESPSA (*p* = 0.01 and 0.02, respectively). Conversely, no significant differences in live/dead ratio were found between groups at 24 h (*p* = 0.10).

## 4. Discussion

Dental implant failure can be the consequence of the host-to-implant tissue response, or of peri-implant microbial infection [20,23,27,28,29,30]. Osteoblast maturation and epithelial sealing can be fostered by several implant topography features [20,31,32]. However, surface characteristics like roughness favor bacterial adhesion, facilitate the formation of complex biofilms and hinder biofilm removal [31,32,33]. Inadequate oral hygiene and an overgrowth of pathobionts, among others, promote the establishment of a dysbiotic microbiota in the peri-implant sulcus, which may result in peri-implant infection [30,31]. Therefore, coating implants with antibacterial and antiadhesion compounds is also a topic of research. Our study used six types of titanium substrates with three different levels of roughness (low, medium and high) with or without TESPSA (Table 1), and a seventh low roughness surface treated with citric acid. TESPSA belongs to the family of silanes, which are compounds used to anchor antibacterial molecules to implants [20]. Furthermore, TESPSA itself possesses antibacterial and osteoinductive properties [19]. Furthermore, exposure to citric acid has shown promising results in the decontamination of titanium implants and in favoring osseointegration [22,34].

The values of roughness and surface free energy (SFE) observed in the M and M-TESPSA surfaces were similar to those of previous studies [19,20,35]. Differences in roughness between L and M surfaces with and without TESPSA were below 0.2 µm, which is considered to produce no differences in bacterial recolonization [23]. Indeed, TESPSA surfaces presented with a lower contact angle and SFE compared to uncoated samples—with these differences being more notoriously observed as roughness increases. These observations may be explained by the fact that a higher roughness may imply a higher surface area, and hence a higher amount of TESPSA molecules adhered to the surface. Moreover, CA did not significantly modify roughness and SFE but did decrease the contact angle of water compared to L (Table 1), as described elsewhere [23,34]. All surfaces were colonized in vitro by eight bacterial species that have been described in the peri-implant sulcus, including primary, secondary and late colonizers [7,35,36]. To observe early colonization, the biofilm was grown for 12 and 24 h [28]. The number of bacteria quantified by qPCR increased with roughness at 12 h, although no significant differences could be observed by either qPCR or CLSM regardless of the presence of TESPSA or citric acid. However, when comparing only coated surfaces at 12 h, H-TESPSA showed a significantly higher number of bacteria than CA (low roughness), and then L-TESPSA and M-TESPSA with a significance lower than 0.1. Thus, among the coated surfaces, biofilm formation was higher when there was greater roughness and SFE, and a lower contact angle (Table 1). This is in accordance with previous studies, which describe the influence of physicochemical implant properties on bacterial adhesion [31,32,37] and cellular re-attachment after disinfection protocols [38]. Bacterial adhesion is facilitated by hydrophilicity (i.e., low contact angle and high SFE), but above all, by roughness [15,23], which also protects against shear forces and decontamination methods [31,33]. It is noteworthy that the results from qPCR, vPCR and CLSM were more homogeneous between surfaces at 24 h, even for the CA and TESPSA surfaces (Figure 3, Figure 4 and Figure 5); the biofilm significantly increased in bacterial load (M and CA; Figure 3), area (L, H-TESPSA and CA; Figure 4) and volume (L-TESPSA and M-TESPSA; Figure 5) from 12 to 24 h; and mortality was lower (though not significantly) at 24 h (Figure 4). Given this time-related trend, differences between surfaces may have been observed by studying initial biofilm formation at earlier times.

The experimental design of this study, especially the complexity of the biofilm model, could explain part of the dissimilarities observed with previous studies. High bacterial adhesion and biofilm formation has been described on rougher surfaces using monospecies models [33,37], but the same surfaces showed no significant differences in biofilm formation at 24 h in vivo [33]. In said assay, due to the heterogeneity of the oral microbiota, different species may cooperate to adhere to the surfaces and once adhered, the process of maturation may overcome differences in roughness [33]. Therefore, the lack of significant differences related to roughness in our study could have resulted from the use of an eight-species mock peri-implant biofilm. Conversely, other in vivo studies have described a higher biofilm formation on rougher surfaces [15]. This highlights the importance of roughness in protecting the biofilm from shear forces and decontamination methods [15,33], events that do not occur in in vitro experiments. Regarding the antibacterial effect of TESPSA, some studies have mentioned a lower bacterial adhesion after 2 h of incubation on titanium surfaces coated with TESPSA [20], and a decreased biofilm formation after 24 h [20] or long-term incubation [19]. However, these assays used monospecies biofilms of *Streptococcus sanguinis* or *Lactobacillus salivarius* [19,20]. Another experimental design based on a more complex biofilm (a six-day five-species community) compared titanium surfaces comprising three degrees of roughness and with or without a silver coating with a L-TESPSA surface [21]. By CLSM, bacterial adhesion and biofilm formation were observed to increase with roughness, observing the lowest adhesion on L-TESPSA. However, these observations may be related to the biofilm structure, since no differences were observed in the number of cells by qPCR [21]. In our study, where surfaces were exposed to an eight-species biofilm, no significant effect of TESPSA was detected by either qPCR or CLSM. The greater number of bacterial phenotypes may have compensated the antibacterial activity. In fact, a higher bacterial diversity may increase the probability of interspecies interactions and the potential formation of coaggregation bridges for each adhered cell, facilitating the biofilm maturation and overcoming differences in roughness and coating [15,33]. Contrary to this idea, the antibacterial effect of TESPSA has been reported in implants exposed to plaque [19]. However, that plaque was extracted from a single volunteer of unreported health status and the microbial composition was unknown, while our model included peri-implant and periodontal pathogens like *F. alocis* and *P. gingivalis* [7,33], which may have contributed beneficial properties for biofilm formation.

The citric acid-passivated surface showed almost no significant differences with the other surfaces, which may also be due to the complexity of the model. A study that exposed titanium surfaces to a single-species model (*S. sanguinis*) found a significant 4.82-fold decrease in CFU/mL in CA [22], but the clinical impact of this reduction is questionable. A similar study used CLSM and SEM to observe a significant reduction in the adhesion of *S. sanguinis* (Gram-positive) but not of *Pseudomonas aeruginosa* (Gram-negative) to titanium passivated with 20% citric acid [23]. This raises the question of whether TESPSA surfaces showing a significant reduction in the bacterial adhesion of *S. sanguinis* and *L. salivarius* monospecies models [19,20] would have produced significant results in Gram-negative bacteria. Other studies proved that, although citric acid failed to prevent implant recolonization, it was effective in killing and removing biofilm pre-formed on implants without causing damage to the oral tissues [23,34], thus this compound is worthy of further research.

In this study, the difference in quantity of bacteria recovered from the different surfaces was lower than 10-fold (Figure 3). From an applied perspective, such a small difference is unlikely to result in an improvement of the clinical outcome; therefore, the choice of any of these surfaces should be based on their capacity to improve the host-to-implant tissue response. Even if bacterial adhesion had been reduced at earlier times (i.e., adhesion was tested at 2 h elsewhere [20]), microbial communities could have compensated differences in implant topography during the maturation of the biofilm [35]. Not having compared the number of bacteria attached to the surfaces at times earlier than 12 h may have been a limitation of this study; however, it was considered not relevant from a clinical perspective.

Within a similar research field, poor results for promising coatings have also been described in toothbrushes, where antibacterials like triclosan have been used as coaters to diminish the bacterial contamination of the bristles [39,40]. Therefore, finding implant surface modifications that have a significant impact on bacterial adhesion and biofilm formation is likely to require further extensive research. Apart from changing the topography and adhering antimicrobials, surface modifications could also include the anchor of biocompatible nanoparticles that could steadily and locally release antibacterial compounds [41]. Moreover, any research should also consider the hard and soft tissue response to the implant to decrease the probability of failure. On the whole, research should focus on promoting the establishment and maintenance of homeostatic peri-implant microbiota. From our study, we conclude that any experimental design on dental implants should be tested on complex in vitro multispecies biofilms or in vivo in experimental human models.

## 5. Conclusions

In the present study, a multispecies oral biofilm model was developed that allowed the determination of the antiadhesion and antimicrobial ineffectiveness of various titanium surfaces. It should be noted, however, that a low adhesion and a high antibacterial effect on the L-TESPSA were observed.

## Figures and Tables

**Figure 1 materials-16-04592-f001:**
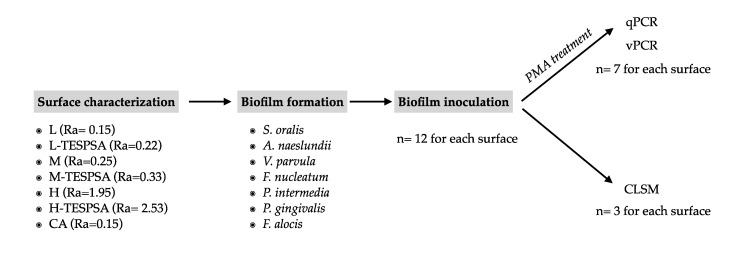
Flowchart of the study.

**Figure 2 materials-16-04592-f002:**
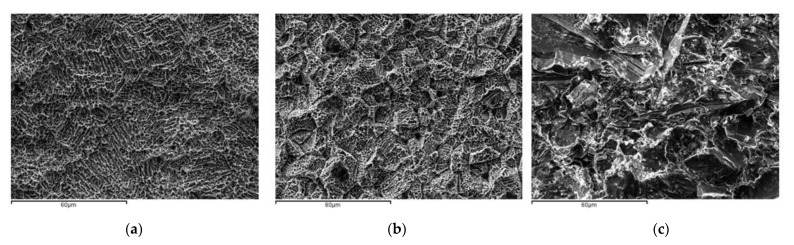
Surfaces observed by SEM. (**a**) Low roughness, (**b**) medium roughness, (**c**) high roughness.

**Figure 3 materials-16-04592-f003:**
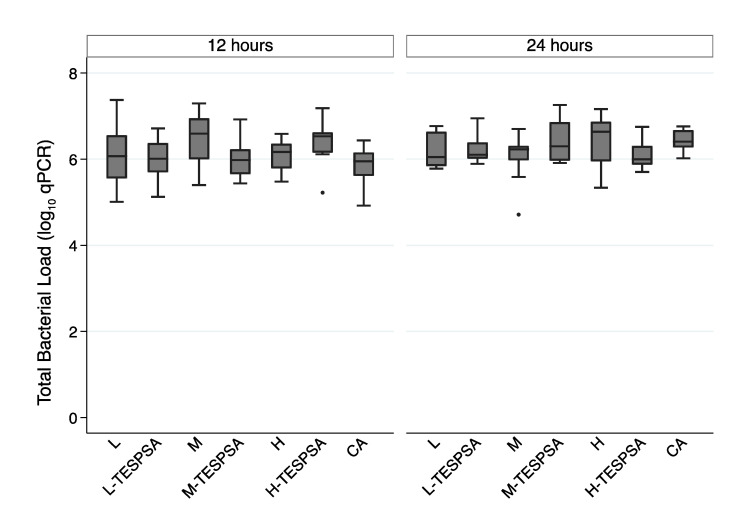
Total bacterial load among different surfaces at 12 and 24 h analyzed by qPCR. L: low; M: medium; H: high; CA: citric acid.

**Figure 4 materials-16-04592-f004:**
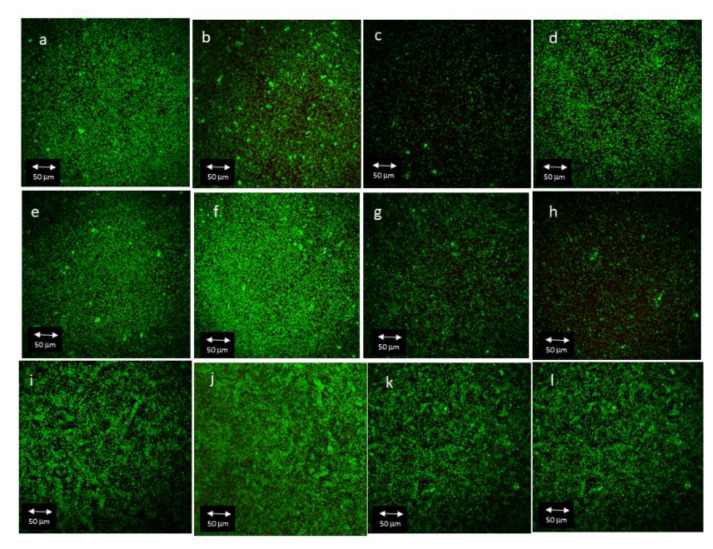
CLSM images from *L* surface at (**a**) 12 and (**b**) 24 h, from L-TESPSA surface at (**c**) 12 and (**d**) 24 h, from M surface at (**e**) 12 and (**f**) 24 h, from M-TESPSA surface at (**g**) 12 and (**h**) 24 h, from H surface at (**i**) 12 and (**j**) 24 h, from H-TESPSA surface at (**k**) 12 and (**l**) 24 h.

**Figure 5 materials-16-04592-f005:**
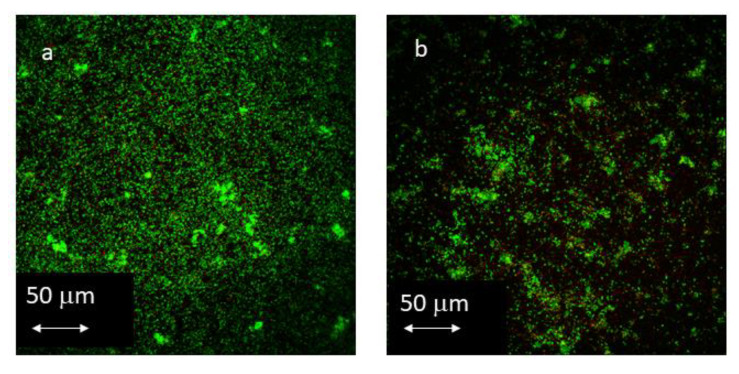
CLSM images from CA surface at (**a**) 12 and (**b**) 24 h.

**Table 1 materials-16-04592-t001:** Description of surface roughness, contact angle and surface energy are shown in mean ± standard deviation. L: low; M: medium; H: high; CA: citric acid.

Surface Name	Roughness(Ra) (μm)	Contact Angle (◦)		Surface Energy (mJ/m^2^)
		H_2_O	Formamide	
L	0.15 ± 0.06	55.4 ± 6	33.6 ± 4	49.6 ± 3
L-TESPSA	0.22 ± 0.02	51.5 ± 2	26.5 ± 3	53.4 ± 2
M	0.25 ± 0.07	58.4 ± 2	34.6 ± 6	48.5 ± 3
M-TESPSA	0.33 ± 0.07	40.4 ± 2	13.0 ± 11	59.2 ± 3
H	1.95 ± 0.19	89.9 ± 10	63.2 ± 10	38.8 ± 14
H-TESPSA	2.53 ± 0.15	20.3 ± 4	0 ± 0	68.3 ± 2
CA	0.15 ± 0.04	51.8 ± 15	37.6 ± 10	49.9 ± 9

**Table 2 materials-16-04592-t002:** *p*-value comparisons of the total bacterial load adhered among antibacterial surfaces measured by qPCR at 12 h.

Group	L-TESPSA	M-TESPSA	H-TESPSA
M-TESPSA	0.403		
H-TESPSA	0.062	0.072	
CA	0.308	0.279	0.021 *

* *p*-value < 0.05.

**Table 3 materials-16-04592-t003:** Biofilm mortality among different surfaces at 12 and 24 h analyzed through the ratio qPCR/vPCR. Values are expressed in log_10_. L: low; M: medium; H: high; CA: citric acid; SD: standard deviation, IQR: interquartile range.

	12 h		24 h	
Group	Mean ± SD	Median (IQR)	Mean ± SD	Median (IQR)
L	0.32 ± 0.25	0.26 (0.38)	0.26 ± 0.22	0.14 (0.29)
L-TESPSA	0.49 ± 0.20	0.42 (0.28)	0.25 ± 0.15	0.22 (0.22)
M	0.43 ± 0.07	0.41 (0.13)	0.36 ± 0.16	0.29 (0.15)
M-TESPSA	0.58 ± 0.16	0.53 (0.25)	0.38 ± 0.24	0.29 (0.08)
H	0.54 ± 0.24	0.52 (0.42)	0.38 ± 0.15	0.29 (0.28)
H-TESPSA	0.44 ± 0.08	0.43 (0.07)	0.28 ± 0.12	0.27 (0.19)
CA	0.61 ± 0.18	0.49 (0.61)	0.39 ± 0.18	0.36 (0.20)

**Table 4 materials-16-04592-t004:** Median bacterial biofilm area (mm^2^) among surfaces at 12 and 24 h analyzed by CLSM. L: low; M: medium; H: high; CA: citric acid. Values expressed in log_10_ as median (IQR).

	12 h		24 h	
Group	Live	Dead	Live	Dead
L	3.40 (0.12)	1.44 (0.49)	3.53 (0.18)	1.8 (0.36)
L-TESPSA	3.43 (0.48)	1.92 (0.54)	3.48 (0.22)	1.58 (0.93)
M	3.44 (0.35)	1.58 (1.25)	3.36 (0.60)	1.52 (0.60)
M-TESPSA	3.30 (0.37)	1.13 (0.49)	3.3 (0.47)	1.55 (0.58)
H	3.13 (0.56)	1.23 (0.31)	3.25 (0.36)	2.36 (0.46)
H-TESPSA	3.09 (0.34)	1.11 (0.09)	3.65 (0.36)	2.12 (0.05)
CA	3.04 (0.37)	1.52 (0.60)	3.51 (0.32)	2.12 (0.76)

**Table 5 materials-16-04592-t005:** Median bacterial biofilm volume (mm^3^) among surfaces at 12 and 24 h analyzed by CLSM. L: low; M: medium; H: high; CA: citric acid. Values expressed in log_10_ as median (IQR).

	12 h		24 h	
Group	Live	Dead	Live	Dead
L	4.83 (0.27)	2.79 (0.28)	5.06 (0.21)	3.36 (0.05)
L-TESPSA	4.71 (0.46)	3.19 (0.42)	4.97 (0.47)	3.45 (0.89)
M	4.81 (0.30)	3.01 (1.11)	4.94 (0.87)	3.15 (0.56)
M-TESPSA	4.58 (0.42)	2.42 (0.54)	4.81 (0.97)	3.23 (0.63)
H	4.58 (0.52)	2.6 (0.28)	4.7 (0.30)	3.68 (0.59)
H-TESPSA	4.48 (0.35)	2.51 (0.02)	5.21 (0.38)	3.74 (0.14)
CA	4.48 (0.53)	2.96 (0.68)	5.14 (0.48)	3.68 (0.76)

**Table 6 materials-16-04592-t006:** Live/dead ratio among different surfaces at 12 and 24 h analyzed by CLSM. Values are expressed in log_10_. L: low; M: medium; H: high; CA: citric acid; SD: standard deviation, IQR: interquartile range.

	12 h		24 h	
Group	Mean ± SD	Median (IQR)	Mean ± SD	Median (IQR)
L	2.45 ± 0.78	2.36 (0.81)	2.02 ± 0.23	1.98 (0.36)
L-TESPSA	1.88 ± 0.38	1.73 (0.67)	2.33 ± 0.71	2.15 (0.81)
M	2.98 ± 3.21	3.12 (2.62)	2.28 ± 0.78	2.31 (1.35)
M-TESPSA	4.72 ± 0.78	3.09 (1.61)	1.99 ± 0.47	2.01 (0.77)
H	2.56 ± 0.40	2.56 (0.29)	1.48 ± 0.27	1.40 (0.46)
H-TESPSA	2.96 ± 0.30	3.06 (0.46)	1.70 ± 0.22	1.66 (0.26)
CA	1.86 ± 0.20	1.84 (0.36)	1.79 ± 0.46	1.73 (0.82)

## Data Availability

The data that support the findings of this study are available from the corresponding author upon reasonable request.

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
