# Peer review of "Bacterial Adhesion of TESPSA and Citric Acid on Different Titanium Surfaces Substrate Roughness: An In Vitro Study with a Multispecies Oral Biofilm Model"

_materials, 2023, doi:10.3390/ma16134592_

Round 1

Reviewer 1 Report

Introduction

Since your article focuses on titanium coatings, that are novel in the dentistry application and differ from the simply surface modification (SLA, MAC, etc.), provide a background on the topic including novel and alternative type of coating. For this propose discuss and cite the following article doi: 10.3390/ma16010246.  

- add the study hypotheses after the aim of the study

Materials and methods

- Specify how many samples were used in the study (n = X) and how many samples were used for each analysis. 

- Add a flowchart of the study at the beginning of the M&M section to simplify the immediate understanding of the workflow you followed.

- Figure 1 doesn't have any link in the text. Furthermore, SEM analysis is not reported in the M&M section. If SEM analysis was performed, report and describe the analysis in details. 

Discussion

- Discuss if the study hypotheses were accepted or rejected based on the results of the study. 

- Since the re-attachment of healthy tissues after the removal of the biofilm is a crucial moment in the treatment of peri-implantitis, discuss the possible effect of different surface roughness in the post treatment cellular attachment. For this propose, discuss and cite a very recently published article that analyzed the biocompatibility of different surface topography and roughness after being exposed to in vitro simulate peri-implantitis. 

Alovisi, M.; Carossa, M.; Mandras, N.; Roana, J.; Costalonga, M.; Cavallo, L.; Pira, E.; Putzu, M.G.; Bosio, D.; Roato, I.; Mussano, F.; Scotti, N. Disinfection and Biocompatibility of Titanium Surfaces Treated with Glycine Powder Airflow and Triple Antibiotic Mixture: An In Vitro Study. Materials 202215, 4850. https://doi.org/10.3390/ma15144850

Author Response

REVIEWER 1

Dear Reviewer,

Thanks for taking the time to review our manuscript and suggest to us to improve our work by providing a lot more detail. We have done so, and we are now submitting a manuscript that not only addresses the points that you specifically raised but also many others that we have considered in order to deliver what we think is a much improved version of our work. This version includes more paragraphs, English grammar revisions in all main sections and new references. Thanks a lot. We are looking forward to your comments.

Sincerely,

Javier Gil

Introduction

Since your article focuses on titanium coatings, that are novel in the dentistry application and differ from the simply surface modification (SLA, MAC, etc.), provide a background on the topic including novel and alternative type of coating. For this propose discuss and cite the following article doi: 10.3390/ma16010246.

Response: We would like to thank the reviewer for his/her input. We have cited the suggested article.  Reference #17.

- add the study hypotheses after the aim of the study

Response: The hypothesis of the study has been added after the aim of the study as:

“The main hypothesis of this study was that antibacterial coatings would have greater antiadhesive and antibacterial effect in machined substrates than in rough ones”

Materials and methods

- Specify how many samples were used in the study (n = X) and how many samples were used for each analysis. 

Response: The number of samples included in the study are stated in line 123. A total of 12 replicates were used for each surface. For qPCR and viability qPCR a total of 7 discs were used, respectively. For CLSM a total of 3 discs were used. It has been detailed in the manuscript and in the flowchart of the study suggested from the reviewer.

- Add a flowchart of the study at the beginning of the M&M section to simplify the immediate understanding of the workflow you followed.

Response: We would like to thank the reviewer for the suggestion. Authors have included a flowchart of the study at the beginning of the M&M section.

- Figure 1 doesn't have any link in the text. Furthermore, SEM analysis is not reported in the M&M section. If SEM analysis was performed, report and describe the analysis in details. 

Response: We are sorry for this issue. We have linked the Figure 1 (in the revised manuscript Figure 2) and detailed the SEM analysis in the M&M section as part of the surface characterization analysis.

Discussion

- Discuss if the study hypotheses were accepted or rejected based on the results of the study. 

Response: We thank the reviewer for this insight. Although there is not a statement of accepting or rejecting the hypothesis, authors believe that the lack of acceptance of the alternative hypothesis has been discussed along the discussion section in different paragraphs such as:

“The number of bacteria quantified by qPCR increased with roughness at 12 h, although no significant differences could be observed by either qPCR or CLSM regardless of the presence of TESPSA or citric acid”

“Therefore, the lack of significant differences related to roughness in our study could have resulted from the use of an 8-species mock peri-implant biofilm.”

“In our study, where surfaces were exposed to an 8-species biofilm, no significant effect of TESPSA was detected by either qPCR or CLSM. The greater number of bacterial phenotypes may have compensated the antibacterial activity. In fact, a higher bacterial diversity may increase the probability of interspecies interactions and the potential formation of coaggregation bridges for each adhered cell, facilitating the biofilm maturation and overcoming differences in roughness and coating [15,30].”

“In this study, the difference in quantity of bacteria recovered from the different surfaces was lower than 10-fold (Figure 2). From an applied perspective, such a small difference is unlikely to result in an improvement of the clinical outcome, and therefore the choice of any of these surfaces should be based on their capacity to improve the host-to-implant tissue response.”

- Since the re-attachment of healthy tissues after the removal of the biofilm is a crucial moment in the treatment of peri-implantitis, discuss the possible effect of different surface roughness in the post treatment cellular attachment. For this propose, discuss and cite a very recently published article that analyzed the biocompatibility of different surface topography and roughness after being exposed to in vitro simulate peri-implantitis. 

Alovisi, M.; Carossa, M.; Mandras, N.; Roana, J.; Costalonga, M.; Cavallo, L.; Pira, E.; Putzu, M.G.; Bosio, D.; Roato, I.; Mussano, F.; Scotti, N. Disinfection and Biocompatibility of Titanium Surfaces Treated with Glycine Powder Airflow and Triple Antibiotic Mixture: An In Vitro Study. Materials 202215, 4850. https://doi.org/10.3390/ma15144850

Response: Thanks to the reviewer for his/her suggestion. We have included the reference in the discussion section. Reference #35.

Reviewer 2 Report

My concern about this manuscript "Early prevention of bacterial adhesion on titanium surfaces of 2 different substrate roughness with antibacterial coating: an in 3 vitro study with a multispecies oral biofilm model" is given below-

Title: Revision required make more simple title 

Abstract: CA was the surface with the lowest bacterial counts and higher mortality at 12 and 24 h, while High harbored  the highest amount of biofilm at 24 h, author justifies this sentence 

By CLSM, CA presented significant amounts of dead cells 22 compared to Medium-TESPSA and High-TESPSA. A significantly greater volume of dead cells was 23 found at 12 hours in Low-TESPSA compared to Medium-TESPSA, while CA also presented signif- 24 icant amounts of dead cells compared to Medium-TESPSA and High-TESPSA.- it's okay but author should represent with material MIC and MBC value or LD 50 dose and Zone inhibition pattern 

A significantly greater volume of dead cells was 23 found at 12 hours in Low-TESPSA compared to Medium-TESPSA, why? justify here 

The introduction need some revision, the author should amend the mechanism of antimicrobial action of materials against pathogen or bacteria

2.2. Bacterial strains and growth conditions 110 The biofilm was composed of eight bacterial species: Streptococcus oralis, Actinomy- 111 ces naeslundii, Veillonella parvula, Fusobacterium nucleatum, Prevotella intermedia, Por- 112 phyromonas gingivalis, Porphyromonas endodontalis and Filifactor alocis, microoganism name should be italics  

The functional coating did not inhibit the colonization of the oral biofilm in the first stages of its development, This indicates that the antimicrobial properties of the developed coatings are not sufficient to inhibit the colonization of oral bacteria. remove this sentence because it is not a conclusive remark and replaced with material synthesis and properties 

It should be noted, however, that a lower adhesion and higher antibacterial effect on the L-TESPSA 382 surface was observed at what concentration or dose and amount?

. Also, CA surfaces presented promising results that suggest that further research should be carried out with this surface but your study support TESPSA promising results in comparison to the CA kindly clarify 

no

Author Response

REVIEWER 2

Dear Reviewer,

Thanks for taking the time to review our manuscript and suggest to us to improve our work by providing a lot more detail. We have done so, and we are now submitting a manuscript that not only addresses the points that you specifically raised but also many others that we have considered in order to deliver what we think is a much improved version of our work. This version includes more paragraphs, English grammar revisions in all main sections and new references. Thanks a lot. We are looking forward to your comments.

Sincerely,

Javier Gil

My concern about this manuscript "Early prevention of bacterial adhesion on titanium surfaces of 2 different substrate roughness with antibacterial coating: an in 3 vitro study with a multispecies oral biofilm model" is given below-

Title: Revision required make more simple title 

Response: Thanks to the reviewer for his/her comment. The title has been simplified to:

“Bacterial adhesion of TESPSA and citric acid on different titanium surfaces substrate roughness: an in vitro study with a multispecies oral biofilm model”

Abstract: CA was the surface with the lowest bacterial counts and higher mortality at 12 and 24 h, while High harbored the highest amount of biofilm at 24 h, author justifies this sentence 

Response: Authors justified these differences in the results taking into consideration the different Ra values from the surfaces. Moreover, it is known that CA possesses antimicrobial activity – which will help to inhibit biofilm adhesion and formation at early stages.

By CLSM, CA presented significant amounts of dead cells 22 compared to Medium-TESPSA and High-TESPSA. A significantly greater volume of dead cells was 23 found at 12 hours in Low-TESPSA compared to Medium-TESPSA, while CA also presented signif- 24 icant amounts of dead cells compared to Medium-TESPSA and High-TESPSA.- it's okay but author should represent with material MIC and MBC value or LD 50 dose and Zone inhibition pattern 

 Response: We are very thankful with the reviewer comment. Under CLSM methodology, it was not possible to assess MIC and MBC. We could only evaluate biofilm mortality in 3D analysis. In the present study, we also used culture technique to evaluate the real mortality of biofilms among different surfaces. The determination of MIC and MBC would require having a suspension with the antimicrobial actives that had been coated over the titanium surfaces. Unfortunately, this was not possible as the mechanism of action evaluated in this study (i.e., biofilm inhibition in antimicrobial coated surfaces) is far away from a killing bacterium in a suspension.

A significantly greater volume of dead cells was 23 found at 12 hours in Low-TESPSA compared to Medium-TESPSA, why? justify here 

Response: We regret not to have a substantial biological rationale for this comment. However, it could be speculated with caution that bacterial biofilm is more labile in machined surfaces than in rough ones. In this context, antimicrobial activity could be increased when the biofilm is less stable.   

The introduction needs some revision, the author should amend the mechanism of antimicrobial action of materials against pathogen or bacteria

Response: Authors have incorporated the mechanism of action of citric acid as follows:

“CA at low pH is able to freely cross the microbial membrane and once inside the cytoplasm, it dissociates into CA anions and protons leading to the acidification of the intracellular media - thus causing functional and structural damage to the bacteria (24).”

2.2. Bacterial strains and growth conditions 110 The biofilm was composed of eight bacterial species: Streptococcus oralis, Actinomy- 111 ces naeslundii, Veillonella parvula, Fusobacterium nucleatum, Prevotella intermedia, Por- 112 phyromonas gingivalis, Porphyromonas endodontalis and Filifactor alocis, microoganism name should be italics  

Response: Thanks to the reviewer. We have italicized the microorganism names accordingly.

The functional coating did not inhibit the colonization of the oral biofilm in the first stages of its development, This indicates that the antimicrobial properties of the developed coatings are not sufficient to inhibit the colonization of oral bacteria. remove this sentence because it is not a conclusive remark and replaced with material synthesis and properties 

Response: Following reviewers’ comment, authors have removed this sentence.

It should be noted, however, that a lower adhesion and higher antibacterial effect on the L-TESPSA 382 surface was observed at what concentration or dose and amount?

Response: As requested by another reviewer, this statement has been adapted as “It should be noted, however, that a low adhesion and high antibacterial effect on the L-TESPSA surface was observed.”  

. Also, CA surfaces presented promising results that suggest that further research should be carried out with this surface but your study support TESPSA promising results in comparison to the CA kindly clarify 

Response: In order to avoid confusion, authors have omitted the last sentence of the discussion and adapted the conclusion as follows:

“In the present study, a multispecies oral biofilm model was developed that allowed determining the antiadhesion and antimicrobial ineffectiveness of various titanium surfaces. It should be noted that a low adhesion and high antibacterial effect on the L-TESPSA and CA surfaces was observed.”

Reviewer 3 Report

1.        Please implement the italic font for the Latin names of microorganisms/in vitro/in vivo throughout the manuscript.

2.        Please provide more information about current use of TESPSA in technology in the introduction section. Discuss the pros and cons.

3.       L48 : what does it means” higher levels of Porphyromonas gingivalis,” Higher than what?

4.       Why have You decided to peform the test on multispecies biofilm instead of single-specie of selected microorganisms?

5.       Biofilm formation: What kind of broth was used to prepare bacterial suspension?

6.       Why the Authors did not use artificial saliva or at least the addition of mucin to the broth? Mucin plays a very important role in biofilm formation especially in adhesion phase.

7.       Table S10 should be in the main manuscript.

8.       Please reorganize this part as the number -18- occurs to many times.

“There-267 fore, coating implants with antibacterial and antiadhesion compounds is also a topic of 268 research 18. Our study used 6 types of titanium substrates with 3 different levels of rough-269 ness (Low, Medium and High) with or without TESPSA (Table 1), and a seventh low 270 roughness surface treated with citric acid. TESPSA belongs to the family of silanes, which 271 are compounds used to anchor antibacterial molecules to implants 18. Besides, TESPSA 272 itself possesses antibacterial and osteoinductive properties 18. Furthermore, exposure to 273 citric acid has shown promising results in the decontamination of titanium implants and 274”

9.       Line 307: “Higher bacterial 307 adhesion and biofilm formation has been described….” Higher than what?

10.    Conclusion: It should be 381 noted, however, that a lower adhesion and higher antibacterial effect on the L-TESPSA 382 surface was observed. Higher than what? Lower than what?

English grammar and style must be improved.

Author Response

REVIEWER 3

Dear Reviewer,

Thanks for taking the time to review our manuscript and suggest to us to improve our work by providing a lot more detail. We have done so, and we are now submitting a manuscript that not only addresses the points that you specifically raised but also many others that we have considered in order to deliver what we think is a much improved version of our work. This version includes more paragraphs, English grammar revisions in all main sections and new references. Thanks a lot. We are looking forward to your comments.

Sincerely,

Javier Gil

  1. Please implement the italic font for the Latin names of microorganisms/in vitro/in vivo throughout the manuscript.

Response: Thanks to the reviewer for the comment. Following the MDPI style guides, the name of microorganisms has been changed to italic font. However, the name of in vitro and in vivo has not been changed as stated in the normative of MDPI journals:

  1. Structure and formatting

3.6 Italics

“Foreign words do not need to be highlighted or italicized, including Greek/Latin terms, such as i.e., e.g., etc., et al., vs., ca., cf., in vivo, ex vivo, in situ, ex situ, in vitro” 

  1. Please provide more information about current use of TESPSA in technology in the introduction section. Discuss the pros and cons.

Response: Done. A new paragraph has been introduced and three news references according to the reviewer comment.         

  1. L48: what does it means” higher levels of Porphyromonas gingivalis,” Higher than what?

Response: It has been corrected as “high”.

  1. Why have you decided to perform the test on multispecies biofilm instead of single-specie of selected microorganisms?

Response: Authors would like to thank the reviewer to bring up this issue. Although our research group have performed studies with single-species biofilm (Godoy-Gallardo et al. 2016, Buxadera et al. 2020), we have more experience with multispecies biofilms (Álvarez et al, Vilarrasa et al. 2018, Blanc et al. 2014). It is well known that multispecies biofilm simulates better intraoral microbial conditions and biofilm formation.

Biofilm formation: What kind of broth was used to prepare bacterial suspension?

Response: The broth used to prepare bacterial suspension was the one used in Alvarez et al. 2013. This suspension is composed of modified brain heart infusion (brain heart infusion broth 37 g/I, mucin from porcine stomach type III [Sigma-Aldrich Chemie GmbH, Buchs, Switzerland] [2.5 g/I], yeast extract [1 g/I], L- cysteine [10.1 g/1), sodium bicarbonate |2 g/l and supplemented with hemin |5 mg/l, menadione [1 mg/I] and glutamic acid [0.25%].

Authors have further detailed this issue in the section 2.3 Biofilm formation.

  1. Why the authors did not use artificial saliva or at least the addition of mucin to the broth? Mucin plays a very important role in biofilm formation especially in adhesion phase.

Response: We could not be more agree with the reviewer in this point. As mentioned in the previous comment, the broth prepared for bacterial suspension contained mucin from porcine stomach type III [Sigma-Aldrich Chemie GmbH, Buchs, Switzerland] [2.5 g/I].

  1. Table S10 should be in the main manuscript.

Response: Following reviewers’ suggestion, we have included S10 as Table 2 in the revised manuscript.

  1. Please reorganize this part as the number -18- occurs to many times.

“There-267 fore, coating implants with antibacterial and antiadhesion compounds is also a topic of 268 research 18. Our study used 6 types of titanium substrates with 3 different levels of rough-269 ness (Low, Medium and High) with or without TESPSA (Table 1), and a seventh low 270 roughness surface treated with citric acid. TESPSA belongs to the family of silanes, which 271 are compounds used to anchor antibacterial molecules to implants 18. Besides, TESPSA 272 itself possesses antibacterial and osteoinductive properties 18. Furthermore, exposure to 273 citric acid has shown promising results in the decontamination of titanium implants and 274”

 Response: Authors have arranged this issue in the revised manuscript as follows:

“Therefore, coating implants with antibacterial and antiadhesion compounds is also a topic of research. Our study used 6 types of titanium substrates with 3 different levels of roughness (Low, Medium and High) with or without TESPSA (Table 1), and a seventh low roughness surface treated with citric acid. TESPSA belongs to the family of silanes, which are compounds used to anchor antibacterial molecules to implants (20). Besides, TESPSA itself possesses antibacterial and osteoinductive properties (19). Furthermore, exposure to citric acid has shown promising results in the decontamination of titanium implants…”

  1. Line 307: “Higher bacterial 307 adhesion and biofilm formation has been described….” Higher than what?

Response: It has been corrected as “high”.

  1. Conclusion: It should be 381 noted, however, that a lower adhesion and higher antibacterial effect on the L-TESPSA 382 surface was observed. Higher than what? Lower than what?

Response: We are sorry for the confusion. It has been corrected as “low” and “high”.

Reviewer 4 Report

The manuscript analyzes the influence of substrate roughness with two bacteriostatic and bactericidal agents on biofilm adhesion by quantifying bacterial adhesion and cell viability.

The authors have submitted quite an interesting manuscript. The topic is interesting to understanding the exact mechanisms of these interactions, but I have found several issues that, once addressed, will improve the manuscript.

General comments

Throughout the text of the manuscript in vitro and in vivo and the bacteria name must be written in italics. 

Materials and Methods

The complexity of the dental biofilm and its formation requires different incubation times for the different bacteria with successive and progressive colonization. Were all eight bacterial species incubated at the same time? I think this aspect can affect the results.

The formation of the acquired pellicle is important for the biofilm formation. Were the discs first incubated with saliva as indicated by several Authors? I think this aspect may affect the results.

Considering the complexity in biofilm formation, I think 12 and 24 h are not enough time to complete the interaction between 8 bacterial species.

What method was used to confirm the total detachment of bacteria from the disc surfaces?

Did the authors examine the quantity of individual bacteria present on different surfaces?

Discussion

The references must be inserted in compliance with the editorial rules.

I think any required changes in Mat and Met section will lead to different considerations in the Discussion sectiom.

Greater attention to the Literature, bearing in mind that several Authors (not mentioned) have dealt with the subject.

33 / 5.000

Risultati della traduzione

Risultato di traduzione

  English needs to be improved

Author Response

REVIEWER 4

Dear Reviewer,

Thanks for taking the time to review our manuscript and suggest to us to improve our work by providing a lot more detail. We have done so, and we are now submitting a manuscript that not only addresses the points that you specifically raised but also many others that we have considered in order to deliver what we think is a much improved version of our work. This version includes more paragraphs, English grammar revisions in all main sections and new references. Thanks a lot. We are looking forward to your comments.

Sincerely,

Javier Gil

The manuscript analyzes the influence of substrate roughness with two bacteriostatic and bactericidal agents on biofilm adhesion by quantifying bacterial adhesion and cell viability.

The authors have submitted quite an interesting manuscript. The topic is interesting to understanding the exact mechanisms of these interactions, but I have found several issues that, once addressed, will improve the manuscript.

General comments

Throughout the text of the manuscript in vitro and in vivo and the bacteria name must be written in italics. 

Response: We would like to thank the reviewer for his/her comment. We did not used an italic format for in vitro and in vivo since we followed the normative of MDPI journals:

  1. Structure and formatting

3.6 Italics

“Foreign words do not need to be highlighted or italicized, including Greek/Latin terms, such as i.e., e.g., etc., et al., vs., ca., cf., in vivo, ex vivo, in situ, ex situ, in vitro” 

Materials and Methods

The complexity of the dental biofilm and its formation requires different incubation times for the different bacteria with successive and progressive colonization. Were all eight bacterial species incubated at the same time? I think this aspect can affect the results.

Response: We would like to thank the reviewer to raise this important issue. This in vitro biofilm growth model is set up so that all species can be inoculated at the same time and appear at the final time of biofilm growth. This is achieved by growing all species to the end of their exponential phase. Each species has a different incubation time. This is key when mixing all the species. These must be inoculated in different concentrations (experimentation previously set up) to achieve a real representation of all of them in the final biofilm. By doing this, it is not necessary to inoculate over time. Below, the methodology is summarized:

“Each bacterial species was grown until they reached the log phase. Each bacterial species needed different time to reach the log phase. The bacterial concentration was adjusted by measuring optical density at 550 nm to obtain bacterial suspensions with concentrations of 105 CFU/ml for each bacterial species, except for S. oralis (104), P. endodontalis (105) and F. alocis (106). The bacterial suspension was composed by the 8 bacterial species.  This bacterial suspension was prepared by diluting overnight pure cultures of each species in fresh modified BHI. Then, each disc was immersed in 1.5 ml of bacterial suspension.”

The formation of the acquired pellicle is important for the biofilm formation. Were the discs first incubated with saliva as indicated by several Authors? I think this aspect may affect the results.

Response: We could not be more agree with the reviewer in this point. The use of saliva has been reported in several in vitro biofilm models to better simulate the oral environment. However, there are also studies from other research groups that did not incubate saliva over the discs and the biofilm was formed (MC Sánchez et al. 2014; MC Sánchez et al 2017). It is worth to mention that the broth used for bacterial suspension contained mucin, which is a protein present in saliva that performs several functions in the oral cavity. Preliminary experiments form this research group have demonstrated that discs pretreatment with sterile clarified human saliva did not provide significant different in adhesion and biofilm thickness when compared to biofilm incubation with BHI plus mucin (data not published).

Considering the complexity in biofilm formation, I think 12 and 24 h are not enough time to complete the interaction between 8 bacterial species.

Response: We would like to thank the reviewer for his/her comment. We also believe that at 12 and 24 hour the biofilm could be “immature”. It could be speculated that some bacteria were at a low concentration due to the short period of study time. Although we did consider to perform 16S for each specific bacteria to be able to quantify each of them, it was discarded for the number of surfaces studied. This is the reason why universal probes were used, as the study intended to assess the overall biofilm adhesion in the early stages of biofilm formation.

Although a complete interaction may be difficult to be achieved in 12 to 24 hours, an study from Fürst et al in 2007 reported that bacterial colonization occurred just 30 minutes after implant installation and that the submucosal microbiota was composed of bacterial species from different clusters (Socrasnky 1998). Below, it could be observed the microbiota that colonizes the implant sites after 30 minutes of implant placement (obtained from Fürst et al. 2007).

What method was used to confirm the total detachment of bacteria from the disc surfaces?

Response: A preliminary internal study evidenced through SEM that 5 minutes of vortexing left practically no cells on the surface of the titanium discs. To corroborate this, we have included SEM images from this assay.

SEM Images from titanium disc without biofilm.

SEM images from titanium disc with biofilm grown

SEM images from titanium discs after vortexing for 5 minutes (where it can be seen that bacterial cells are detached from the titanium surface)

Did the authors examine the quantity of individual bacteria present on different surfaces?

Response:  We would like to thank the reviewer for raising up this issue. The quantity of each individual bacteria was not examined since universal oligonucleotides and universal probes were used. Although we did consider to perform 16S for each specific bacteria to be able to quantify each of them, it was discarded for the number of surfaces studied. This is the reason why universal probes were used, as the study intended to assess the overall biofilm adhesion in the early stages of biofilm formation

Discussion

The references must be inserted in compliance with the editorial rules.

I think any required changes in Mat and Met section will lead to different considerations in the Discussion sectiom.

Greater attention to the Literature, bearing in mind that several Authors (not mentioned) have dealt with the subject.

Response: Thanks to the reviewer. We have adapted his/her comments along the manuscript accordingly.

Round 2

Reviewer 1 Report

The authors revised the manuscript correctly.

Reviewer 2 Report

na

Reviewer 4 Report

I think the work can be accepted. Thanks to the authors for the comments.

Minor editing of English language required